# Astaxanthin Inhibits Diabetes-Triggered Periodontal Destruction, Ameliorates Oxidative Complications in STZ-Injected Mice, and Recovers Nrf2-Dependent Antioxidant System

**DOI:** 10.3390/nu13103575

**Published:** 2021-10-12

**Authors:** Govinda Bhattarai, Han-Sol So, Thi Thu Trang Kieu, Sung-Ho Kook, Jeong-Chae Lee, Young-Mi Jeon

**Affiliations:** 1Cluster for Craniofacial Development and Regeneration Research, School of Dentistry, Jeonbuk National University, Jeonju 54896, Korea; govinda@jbnu.ac.kr; 2Institute of Oral Biosciences, School of Dentistry, Jeonbuk National University, Jeonju 54896, Korea; 3Research Center of Bioactive Materials, Department of Bioactive Material Sciences, Jeonbuk National University, Jeonju 54896, Korea; dusrh12@naver.com (H.-S.S.); trangkt0327@gmail.com (T.T.T.K.); 4Research Institute of Clinical Medicine of Jeonbuk National University, Jeonju 54907, Korea

**Keywords:** astaxanthin, diabetes, periodontal destruction, oxidative damage, hyperglycemic complication, Nrf2 pathway

## Abstract

Numerous studies highlight that astaxanthin (ASTX) ameliorates hyperglycemic condition and hyperglycemia-associated chronic complications. While periodontitis and periodontic tissue degradation are also triggered under chronic hyperglycemia, the roles of ASTX on diabetes-associated periodontal destruction and the related mechanisms therein are not yet fully understood. Here, we explored the impacts of supplemental ASTX on periodontal destruction and systemic complications in type I diabetic mice. To induce diabetes, C57BL/6 mice received a single intraperitoneal injection of streptozotocin (STZ; 150 mg/kg), and the hyperglycemic mice were orally administered with ASTX (12.5 mg/kg) (STZ+ASTX group) or vehicle only (STZ group) daily for 60 days. Supplemental ASTX did not improve hyperglycemic condition, but ameliorated excessive water and feed consumptions and lethality in STZ-induced diabetic mice. Compared with the non-diabetic and STZ+ASTX groups, the STZ group exhibited severe periodontal destruction. Oral gavage with ASTX inhibited osteoclastic formation and the expression of receptor activator of nuclear factor (NF)-κB ligand, 8-OHdG, γ-H2AX, cyclooxygenase 2, and interleukin-1β in the periodontium of STZ-injected mice. Supplemental ASTX not only increased the levels of nuclear factor erythroid 2-related factor 2 (Nrf2) and osteogenic transcription factors in the periodontium, but also recovered circulating lymphocytes and endogenous antioxidant enzyme activity in the blood of STZ-injected mice. Furthermore, the addition of ASTX blocked advanced glycation end products-induced oxidative stress and growth inhibition in human-derived periodontal ligament cells by upregulating the Nrf2 pathway. Together, our results suggest that ASTX does not directly improve hyperglycemia, but ameliorates hyperglycemia-triggered periodontal destruction and oxidative systemic complications in type I diabetes.

## 1. Introduction

Diabetes mellitus (DM) is a metabolic disease characterized by chronic hyperglycemia [1]. Disorders in secretion, action, or both, of insulin induce hyperglycemia, and uncontrolled hyperglycemic condition causes various oxidative and inflammatory complications in the body [2]. DM also promotes bone fragility and disturbs the processes required for defected bone healing [3,4]. Moreover, DM patients show greater periodontal degradation than non-diabetic healthy persons at old ages [5,6,7,8]. These reports indicate that in addition to systemic healthy complications, chronic hyperglycemia triggers inflammatory responses and alveolar bone loss in the periodontium.

Oxidative stress is the hallmark in the development of DM and DM-related complications [9,10]. In general, oxidative stress is induced by excessive production of reactive oxygen species (ROS), impairments in antioxidant defense systems, or both in the body. Persistent and prolonged oxidative stress alters various biological processes, damages cellular macromolecules, and triggers inflammatory and degenerative diseases [11]. High glucose level in the blood facilitates cellular ROS generation and inflammatory responses, by which insulin secretion and its action in insulin-responsive cells are diminished [12]. Chronic inflammatory responses also contribute to a mild and prolonged production of ROS leading to oxidative damages to cells and tissues [13]. Similarly, periodontitis, a chronic periodontal inflammatory disease, involves an excessive ROS generation in the periodontium [14,15]. Therefore, oxidative stress might be the key mediator in DM-associated periodontal destruction, as well as in hyperglycemia-related systemic disorders.

Based on the roles of oxidative stress on inflammatory responses and degenerative diseases, clinical use of non-toxic and antioxidative dietary nutrients is the attractive approach. Numerous studies have demonstrated biological, medicinal, and pharmacological activities of naturally occurring antioxidants such as carotenoids, phenolic acids, and flavonoids. Especially, astaxanthin (3,3′-dihydroxy-β,β′-carotene-4,4′-dione; ASTX) exhibits stronger antioxidant activity than other types of carotenoids [16]. Studies have shown that ASTX treatment ameliorates DM complications via ROS scavenging [16], anti-inflammatory action [17,18], activation of antioxidant defense systems [19,20], and upregulation of nuclear factor erythroid 2-related factor 2 (Nrf2) [21]. ASTX also exhibited protective roles on cancer, aging, and the development of degenerative pathologies in various organs [16]. These reports suggest that ASTX exerts preventive and/or attenuating potentials on the development of DM and DM-associated inflammatory complications.

Even though many investigators highlight the beneficial impacts of ASTX on diabetes or diabetes-associated chronic complications, its role on hyperglycemia-triggered periodontal destruction is not yet completely understood. Here, we evaluated whether supplemental ASTX protects diabetes-triggered destruction of the periodontium, and this effect is directly associated with its potency to ameliorate hyperglycemic condition and/or hyperglycemia-associated systemic complications. To this end, we used the streptozotocin (STZ)-induced diabetic mice that received oral gavage with ASTX or vehicle solution daily for 60 days after STZ injection. To further verify the roles of ASTX, we also examined the effects of ASTX on oxidative stress, osteoblastic differentiation, and proliferation in human periodontal ligament cells (hPDLCs) that were exposed to advanced glycation end products (AGE) in the presence and absence of Nrf2-specific inhibitor. Our current findings demonstrate that oral supplementation with ASTX does not directly reduce high level of blood glucose, but delays body weight loss at early stage after STZ injection together with improvements in water and feed consumptions and survival rate in the diabetic mice. Our results also highlight that supplemental ASTX protects mice from diabetes-triggered periodontal destruction by inhibiting oxidative damage, inflammatory response, and osteoclastic activation, as well as by increasing the expression of osteogenic and antioxidative transcription factors in STZ-injected mice. Overall, our study supports that the protective effect of ASTX on diabetes-associated periodontal disorders is mainly due to its antioxidant and anti-inflammatory potentials via the upregulation of Nrf2 and Nrf2-dependent antioxidant system.

## 2. Materials and Methods

### 2.1. Chemicals and Laboratory Wares

ASTX (CAS No. 472-61-7) and STZ (CAS No. 18883-66-4) were obtained from Sigma-Aldrich Co. LLC (St. Louis, MI, USA). ML385 (CAS No. 846557-71-9/Nrf2-specific inhibitor) and fetal bovine serum (FBS) were obtained from Tocris Biosciences (Avonmouth, Bristol, UK) and HyClone Laboratories, Inc. (Logan, UT, USA), respectively. Antibodies specific to cyclooxygenase-2 (COX-2; BS1076), Nrf2 (BS1258), and runt-related transcription factor 2 (RUNX2, BS2831) were purchased from Bioworld Technology, Inc. (St. Louis Park, MN, USA). Receptor activator of nuclear factor (NF)-κB ligand (RANKL; ALX-804-243) was purchased from Enzo Life Sciences, Inc. (Farmingdale, NY, USA), while 2′,7′-dichlorodihydrofluorescein-diacetate (DCFH-DA), osterix (ab209484), osteopontin (ab8448), and γ-H2AX (ab26350) were from Abcam (Cambridge, UK). The 8-hydroxy-2′-deoxyguanosine (8-OHdG; sc-66036) and antibodies specific to cathepsin K (CTSK; sc-48353), interleukin (IL)-1β (sc-52012), and β-actin (sc-47778) were provided from Santa Cruz Biotechnology (Santa Cruz, CA, USA). AGE was offered from BioVision (Milpitas, CA, USA). Unless otherwise specified, chemicals were purchased from Sigma-Aldrich Co. LLC, while laboratory consumables were from Falcon Labware (Becton-Dickinson, Franklin Lakes, NJ, USA).

### 2.2. Animals, Treatment, and Sample Preparation

C57BL/6 male mice (9 weeks old) were supplied from Orient Bio (Daejeon, Korea) and randomly divided into cages (*n* = 5/cage). All mice were acclimatized to the new laboratory environment 7 days before use. During the experimental period, mice were housed at 22 ± 1 °C and 55 ± 5% humidity on 12 h light/dark auto-cycling with free access to food and water. The conditions of mice were monitored every 12 h per day. Mice were assigned into three groups: non-diabetic control, STZ, and STZ + ASTX groups. Mice in the STZ and STZ-ASTX groups received a single intraperitoneal injection of 150 mg/kg STZ freshly dissolved in 10 mM sodium citrate buffer 6 h after fasting, while non-diabetic mice (control group) were injected with the buffer only. Blood glucose level in mice groups were measured using digital glucometer (Accu-check^®^ Advantage, Roche Diagnostic, Indianapolis, IN, USA) 72 h after STZ injection. Hyperglycemic mice exhibiting the blood glucose level higher than 300 mg/dL were used as the STZ and STZ + ASTX groups. The STZ + ASTX group received 12.5 mg/kg ASTX via oral gavage with 100 µL olive oil (vehicle) daily for 60 days, while the STZ group received the vehicle only at the same time throughout the period. Here, we selected olive oil as a vehicle and the dosage of ASTX considering the experimental designs previously reported [22,23]. The amount of supplemental ASTX was adjusted in regard to the changes in body weight of mice. The diabetes-associated characteristic phenotypes such as blood glucose level, body weight change, and survival rate were monitored in all groups every 10 days after STZ injection. To measure water and feed consumptions, mice in all groups were temporary single-housed for 24 h at 30 days post-hyperglycemia induction in a cage that was equipped with two plastic plates containing feed (10 g) and water (40 mL), respectively. The samples including periodontal tissue, whole blood, pancreas, and bone marrow-derived osteoclastogenic cells were collected 12 h after the final administration of ASTX and used for further analyses.

### 2.3. Micro-Computed Tomography (µCT) Analysis

Alveolar bones were scanned using a desktop scanner (1076 Skyscan Micro-CT; Skyscan, Kontich, Belgium) followed by analysis using CTAn software (Skyscan). The X-ray source was set at 100 kV and 100 µA with a pixel size of 18 µm, a 1 mm filter, and a tomographic rotation of 180° (rotation step of 0.6°). The image slices were reconstructed using a cone-beam reconstruction software based on the Feldkamp algorithm (Dataviewer; Skyscan, Kontich, Belgium). The volumetric measurements of bone volume percentage (BV/TV; %) and bone mineral density (BMD; g/cm^3^) in alveolar bones were performed as described previously [15]. In this analysis, the regions of interest (ROI) were extended from the mesial root of the first molar to the distal root of the third molar, which served as the endpoint landmark borders. The delimited landmark borders and the contours of the ROI were drawn at regular intervals with a slice-based method for 173 slices. The entire bone area of the interproximal and furcation areas was included in the ROI.

### 2.4. H&E Staining of Periodontal and Pancreatic Tissue Samples

The pancreas and decalcified periodontal tissue were fixed in paraformaldehyde, dehydrated, sectioned at a thickness of 5.0 µm, and embedded in paraffin following the methods described previously [15,24]. Tissue sections were stained with hematoxylin and eosin (H&E) before being mounted on glass slides, and images were taken using an optical microscope (Leica DM2500, Wétzlar, Germany) linked with a camera and image processing software (Leica application suite V4). The distance between the cementoenamel junction (CEJ) and the alveolar bone crest (ABC) in periodontal tissue sections was measured in the interproximal regions between the first to second molars and the second to third molars, respectively, using ImageJ program. The islets of Langerhans in pancreatic tissue samples were also observed using the optical microscope (Leica) after H&E staining.

### 2.5. Measurement of Osteoclasts Formed in the Periodontium

Osteoclasts in periodontal tissues were quantified by staining the tissue sections with tartrate-resistant acid phosphatase (TRAP) using a leukocyte acid phosphatase kit (Cosmo Bio, Tokyo, Japan) according to the manufacturer’s instructions. After counterstaining with hematoxylin, the TRAP-positive osteoclasts that were found in and around alveolar bone at the interproximal region including the first to the third molars were counted under the light microscope (Leica, Wétzlar, Germany).

### 2.6. Immunohistochemistry (IHC)

Levels of various signaling molecules specific to osteoclastic (RANKL and CTSK) and osteoblastic activation (RUNX2 and osterix), pre-inflammatory response (COX-2 and IL-1β), oxidative damage (8-OHdG and γ-H2AX), and cellular redox regulation (Nrf2) in periodontal tissue sections were evaluated by IHC. Briefly, tissue sections were stained with each primary antibody (1:200–400 dilutions) specific to the molecules, and their expression levels were determined using rabbit-anti- or mouse-anti-Vectastain ABC DAB-HRP kits (Vector Laboratories, Burlingame, CA, USA) following the manufacturer’s instructions. Expression patterns of the molecules at maxillary interproximal region including the first to the third molars were observed using the light microscope (Leica, Wétzlar, Germany).

### 2.7. Measurements of Serum RANKL Level and Antioxidant Enzyme Activity

Blood serum was separated by centrifuging whole blood samples at 1500 g in a serum separator tube. The level of RANKL in the sera was measured by enzyme-linked immunosorbent assay (ELISA) using a mouse-anti-RANKL ELISA Kit (Abcam). The activities of superoxide dismutase (SOD) and catalase in the blood serum were evaluated using their specific assay kits, No. 706002 for SOD (Cayman Chemical, Ann Arbor, MI, USA) and ECAT-100 for catalase (BioAssay Systems, Hayward, CA, USA), respectively. All procedures followed the manufacturer’s instructions, and enzyme activities were calculated by measuring the absorbance of ROS-sensitive dye using a microplate reader (SPECTROstar^®^ Nano, BMG LABTECH, Ortenberg, Germany).

### 2.8. Isolation of BM Cells from Mice Groups and Osteoclastogenic Assay

Whole BM cells were isolated from femurs and tibias of mice 12 h after the final administration of ASTX. Cells were resuspended in α-minimum essential media (α-MEM, Thermo Fisher Scientific, Waltham, MA, USA) and centrifuged at 2000× *g* for 3 min. The pellets were seeded onto 60 mm tissue culture plates and incubated in α-MEM supplemented with 2 mM glutamine, antibiotics (100 IU/mL penicillin G and 100 μg/mL streptomycin), and 20% FBS. After 24 h of incubation, the supernatants (non-adherent cells) were collected and centrifuged at 2000× *g* for 5 min. The pellets were resuspended and divided into 12-well culture plates (2 × 10^6^ cells/well) in α-MEM supplemented with 5% FBS, 30 ng/mL monocyte-colony stimulating factor (M-CSF), and 50 ng/mL RANKL followed by additional incubation for 7 days. During the incubation, the medium was replaced with fresh medium of the same batch every 3 days. At the end of incubation, the cells were stained with TRAP, and the TRAP-positive osteoclasts were counted using the light microscope (Leica, Wétzlar, Germany).

### 2.9. Measurements of Circulating Lymphocytes and Red Blood Cells (RBC)

Blood samples were collected from mice groups 12 h after the final gavage with ASTX, and levels of circulating lymphocytes and RBC were analyzed using an automated blood cell counter (Sysmex XE-2100; TOA Medical Electronics Co., Kobe, Japan).

### 2.10. In Vitro Assays Using hPDLCs

#### 2.10.1. The hPDLC Culture and Treatment

The hPDLCs were collected from the teeth extracted from healthy patients (18–23 years old) undergoing orthodontic treatment at Jeonbuk National University Dental Hospital (Jeonju, Korea). All donors provided written informed consent for use of their tissues, and this study was approved by the Ethical Committee of the Jeonbuk National University Hospital. Isolation and culture of hPDLCs followed the procedures described previously [25]. The hPDLCs were initially cultured in a growth medium (α-MEM supplemented with 10% FBS, 100 IU/mL penicillin G, and 100 μg/mL streptomycin) in 100 mm culture dish at 37 °C in a humidified atmosphere of 5% CO_2_. The cultures of hPDLCs at the second or third passages were divided onto 60 mm dish (2 × 10^6^ cells/well), 12-well (1 × 10^6^ cells/well), or 96-multiwell culture plates (1 × 10^5^ cells/well) in growth medium. After 12 h of incubation, the culture media were switched to 1% FBS-supplemented α-MEM with and without AGE (200 μg/mL), ASTX (30 or 60 μM), ML385 (5 μM), or all of them. After 48 h of incubation, cellular ROS and superoxide anion levels, Nrf2 induction, and proliferation rate in the hPDLCs were evaluated. Alternatively, parts of the hPDLCs were incubated in osteogenic medium supplemented with 5% FBS, 100 nM dexamethasone, 50 µM of ascorbic acid, and 10 mM β-glycerophosphate (DAG) in the presence and absence of AGE, ASTX, or both. After 5 days of incubation, protein levels of RUNX2 and osteopontin (OPN) in the cells grown on 60 mm dishes were analyzed by immunoblot assay, while mineralization of the cells grown on 12-well culture plates was evaluated by Alizarin Red staining 14 days after incubation.

#### 2.10.2. DCFH-DA Staining and Flow Cytometric Assay

Intracellular ROS accumulation in hPDLCs was determined by measuring the level of DCF, a fluorescent dye derived from the ROS-sensitive DCFH-DA. To this end, DCFH-DA (10 µM) was added to hPDLC cultures at the end of incubation, and after 20 min of additional incubation, DCF-specific signal in the cells (10,000 events/sample) was measured using a FACS Calibur^®^ system (Becton-Dickinson, Franklin Lakes, NJ, USA).

#### 2.10.3. Immunofluorescence Assay

Immunofluorescence assay was carried out to determine level of mitochondrial superoxide anions. Briefly, 5 µM of MitoSOX^TM^ Red (M36008; Invitrogen, Waltham, MA, USA) was added into the hPDLC cultures at the end of incubation allowing an additional maintenance for 15 min at 37 °C. After counterstaining with 4′,6-diamidino-2-phenylindole (DAPI) for 10 min, levels of MitoSOX^TM^ Red-specific fluorescence in the cells were observed under a fluorescence microscope (Leica DMI-4000B, Wétzlar, Germany). Mean fluorescent intensity (MFI) of the red dye in the cells was also calculated using ImageJ program (National Institutes of Health, Bethesda, MD, USA).

#### 2.10.4. Immunoblot Assay

At the end of incubation in the growth or osteogenic medium, the hPDCLs were processed for the isolation of whole protein lysates, and the lysates (20 μg/sample) were separated by sodium dodecyl sulfate-polyacrylamide gel electrophoresis on 10–12% gels followed by electroblotting onto polyvinylidene difluoride membrane. The blots were washed with a buffer (10 mM Tris-HCl (pH 7.6), 150 mM NaCl, and 0.05% Tween-20), blocked with 5% skim milk for 1 h, and incubated with primary Nrf2, RUNX2, OPN, or β-actin antibody (1:1000 or 2000 dilution). The blots were exposed to horseradish peroxidase-conjugated secondary antibody, and immunoreactive bands were visualized by enhanced chemiluminescence (ELPIS-Biotech, Daejeon, Korea) followed by exposure to X-ray film (Eastman Kodak, Rochester, NY, USA).

#### 2.10.5. Proliferation Assay

The proliferation rate of hPDLCs was assessed using Cell Counting Kit-8 (CCK-8; Dojindo Lab, Rockville, MD, USA) at the end of 48 h incubation. Optical density specific to the CCK-8 dye was measured at 450 nm using a microplate reader (SPECTROstar^®^ Nano).

#### 2.10.6. Mineralization Assay

The hPDLCs were incubated in the DAG-supplemented osteogenic medium with and without AGE, ASTX, or both for 14 days. The media were changed every two days throughout the incubation period. At the end of incubation, the degree of mineralization in the cells was determined by staining with Alizarin Red S after fixation for 30 min in 4% paraformaldehyde. The cells stained with the red dye were observed under a light microscope, and the amount of red dye was also quantified by measuring the dye-specific absorbance at 570 nm after treatment with 10% acetylpyridinum chloride.

### 2.11. Statistical Analyses

Unless otherwise specified, all results are expressed as the mean ± standard deviation. One-way ANOVA was applied to find out significant differences among the experimental groups using the ORIGIN 7.0 program (MicroCal software, Northampton, MA, USA). The post-hoc Tukey test was used to determine the significance of differences among the groups. The significance of differences between the two sets of data was also evaluated by two-tailed Student’s *t*-test. A value of *p* < 0.05 was considered statistically significant.

## 3. Results

### 3.1. Oral Supplementation with ASTX Does Not Attenuate Blood Glucose Level, but Improves Water and Feed Consumption and Survival Rate in STZ-Induced Diabetic Mice

A schematic diagram of the experimental designs is shown in Figure 1A. Long-term oral gavage of ASTX did not exert any significant effect on blood glucose level in STZ-induced diabetic mice throughout the experimental period (Figure 1B). The ASTX treatment also did not protect mice from STZ-induced destruction of pancreatic islets (Appendix A). Both the STZ and STZ + ASTX groups showed lower body weights compared with that of control group in a time-dependent manner after STZ injection, whereas the STZ group revealed a relatively acute loss of body weight than did the STZ + ASTX group (Figure 1C). Compared with the control group, the STZ and STZ + ASTX groups exhibited characteristic phenotypes of diabetes such as polydipsia and polyuria, whereas the amounts of water (Figure 1D) and feed consumption (Figure 1E) in the STZ + ASTX group were significantly (*p* < 0.05) less than that in the STZ group 30 days after STZ injection. These results were also similar to when such amounts were determined 60 days post-STZ injection (data not shown). In addition, the STZ + ASTX group revealed a higher survival rate (%) than that of the STZ group 60 days after the hyperglycemia induction (Figure 1F). These results indicate that supplemental ASTX does not directly improve pancreatic damage and hyperglycemic condition, but delays or ameliorates an acute systemic complication in STZ-induced diabetic mice.

### 3.2. Supplemental ASTX Protects Mice from Diabetes-Triggered Periodontal Destruction in STZ-Injected Mice

As the hyperglycemic condition contributes to periodontitis and periodontitis-induced tissue degradation, we explored whether STZ-induced diabetes actually triggers periodontal destruction, and this is suppressed by ASTX supplementation. Compared with the control or STZ + ASTX group, the STZ group revealed severe destruction of alveolar bone followed by the visible exposure of root surface in the furcation region of molars both in the maxilla and the mandible (Figure 2A). The reconstructed 3D images supported greater alveolar bone loss both at the lingual (Figure 2B) and buccal sides (Appendix A) of the maxilla and the mandible in STZ group compared with control or STZ + ASTX group. The STZ group also exhibited significantly lower values of alveolar bone percentage (BV/TV; %) in maxillary and mandibular regions than did the control or STZ+ASTX group (Figure 2C). The STZ-induced alveolar bone loss and its protection by supplemental ASTX were confirmed by determining the value of BMD (g/cm^3^), in which BMD values in the maxilla and the mandible of the STZ + ASTX group were comparable with that of the control group (Figure 2D). The results from H&E staining also highlighted the potency of ASTX to inhibit diabetes-stimulated alveolar bone degradation; the STZ + ASTX group represented almost normal levels at ABC, gingival epithelium, and PDL at the maxilla, whereas the STZ group exhibited severe absorption of alveolar bone along with its replacement to connective tissue (Figure 3A). This was similar to when tissue sections of the mandible were analyzed by H&E staining (data not shown). When the distances between CEJ and ABC in the first to the second molars (Figure 3B) and in the second to the third molars (Figure 3C) of the maxilla were determined, the STZ group represented longer distances than did the control or STZ + ASTX group. Together, these results support that hyperglycemia triggers periodontal destruction in STZ-induced diabetic mice, and this is mostly recovered by long-term supplementation with ASTX.

### 3.3. ASTX Diminishes Osteoclast Formation and the Induction of Osteoclastogenic Factors, but Restores Osteogenic Factor Expression in the Periodontium of STZ-Injected Mice

Preferable activation toward osteoclastic differentiation and/or decreases in osteoblastic activation may cause an imbalanced bone metabolism, eventually leading to excessive bone resorption. Therefore, we evaluated whether the diabetes-triggered alveolar bone loss is due to an imbalanced activation between osteoclasts and osteoblasts. The STZ group showed greater regions stained with TRAP at and around the PDL located between the first to the third molars of the maxilla compared with control or STZ + ASTX group (Figure 4A). The STZ group also exhibited significantly higher numbers of osteoclasts formed at the regions of periodontal tissues than did control (*p* < 0.001) or STZ + ASTX group (*p* < 0.05) (Figure 4B). In addition, the results from IHC assay revealed that the expression levels of RANKL (Figure 4C) and CTSK (Appendix A) in the periodontium of STZ group were apparently greater than those in the control or STZ + ASTX group. Similarly, the level of blood serum-derived RANKL in the STZ group was significantly higher than that in the control (*p* < 0.001) and STZ + ASTX groups (*p* < 0.01) (Figure 4D). When the non-adherent bone marrow cells derived from mice groups were incubated in osteoclastogenic medium, the cells originated from the STZ group showed greater osteoclast forming activity than did the cells from the control (*p* < 0.001) or STZ + ASTX group (*p* < 0.05) (Appendix A). In contrast, the expression levels of RUNX2 (Figure 4E) and osterix (Figure 4F) in periodontal tissues of the STZ group were transparently lower than those in the control or STZ + ASTX group. These results indicate that a local and systemic stimulation for osteoclastic activation along with impaired osteogenic activation in the periodontium occurs under STZ-induced hyperglycemia, whereas this impairment is markedly recovered by supplemental ASTX.

### 3.4. ASTX Attenuates STZ-Induced Oxidative Damage and Inflammatory Response in the Periodontium of Diabetic Mice

While hyperglycemia causes oxidative stress and inflammatory activation, inflammatory response itself also stimulates cellular ROS generation and accumulation [2,16]. To further explore the possible mechanisms by which ASTX protects diabetes-triggered periodontal destruction, we determined the expression levels of several markers corresponding to oxidative stress, DNA damage, and inflammation in periodontal tissue samples by IHC. The STZ group exhibited higher expression of 8-OHdG, an oxidative stress biomarker (Figure 5A), and γ-H2AX, a DNA damage marker (Figure 5B), in the periodontium than did the control or STZ-ASTX group. Compared with the control and STZ + ASTX groups, the STZ group also showed greater expression levels of IL-1β (Figure 5C) and COX-2 (Appendix A), in the periodontal tissues. These results suggest that in addition to the imbalanced bone metabolism, the diabetes-triggered periodontal destruction is orchestrated with the increases in oxidative damage and inflammatory responses in the periodontium.

### 3.5. Supplemental ASTX Restores Levels of Periodontal Nrf2, Serum SOD and Catalase, and Circulating Lymphocytes in STZ-Injected Mice up to Those of Control Mice

We next explored whether ASTX-mediated protection on diabetes-triggered periodontal destruction is due to a recovery of antioxidant defense systems in STZ-injected mice. To this end, we initially compared the level of Nrf2, a master transcriptional regulator for cellular antioxidant systems [21], in the periodontal tissues of mice groups. While the Nrf2 level in the STZ group was barely found, the STZ + ASTX group revealed a marked Nrf2 expression similar to the control group (Figure 6A). The STZ group also showed significantly lower activities of SOD and catalase in the blood serum compared with the control (*p* < 0.01) or STZ + ASTX group (*p* < 0.05) (Figure 6B). Moreover, STZ injection obviously reduced the levels of circulating lymphocytes, but not RBC, at 60 days post-hyperglycemia induction, and this reduction was almost completely recovered by supplemental ASTX (Figure 6C). These results postulate that supplemental ASTX ameliorates hyperglycemia-associated systemic complications by upregulating the Nrf2 and Nrf2-dependent antioxidant system, as well as by recovering an impaired hematopoietic development.

### 3.6. ASTX Treatment Inhibits ROS Accumulation and Recovers Mineralization in AGE-Exposed hPDLCs

We further evaluated the role of ASTX on oxidative stress of hPDLCs exposed to the glycated product, AGE, as a diabetic biomarker. AGE treatment significantly (*p* < 0.001) increased DCF-positive cells (%) in the hPDLCs compared with non-treated control hPDLCs, and this increase was diminished by combined treatment with ASTX in a dose-dependent manner (Figure 7A,B). Immunofluorescence assay also indicated an AGE-stimulated increase in mitochondrial superoxide anions in hPDLCs and its suppression by the direct addition of ASTX (Figure 7C). When the value of MitoSOX^TM^-specific MFI in the hPDLCs was evaluated, exposure to AGE alone markedly increased MFI levels in the cells, and this increase was mostly prevented in combination with 60 μM ASTX (Figure 7D). The addition of AGE significantly (*p* < 0.01) diminished mineralization in DAG-treated hPDLCs, and this reduction was also prevented by the addition of ASTX in a dose-dependent manner (Figure 7E,F). Moreover, immunoblot assay revealed that the ASTX-induced recovery of mineralization in AGE-exposed hPDLCs was accompanied by the restoration of osteogenic markers, RUNX2 and OPN (Figure 7G–I). Together, these results support a direct potency of ASTX itself to suppress cellular oxidative stress and to increase osteogenic differentiation.

### 3.7. ASTX Exerts Its Potentials on Oxidative Stress and Proliferation in AGE-Treated hPDLCs via the Upregulation of Nrf2

As numerous studies highlight that the Nrf2 pathway is the main therapeutic target of ASTX, we explored the role of Nrf2 on ASTX-induced antioxidation in AGE-exposed hPDLCs. The addition of AGE alone did not increase Nrf2 level in hPDLCs, but the incubation in combination with ASTX greatly stimulated the induction of Nrf2 in the cells (Figure 8A,B). ASTX treatment alone also significantly increased (*p* < 0.01) the protein level of Nrf2 in the cells compared with control cells or the cells exposed to AGE alone (Figure 8B). Pretreatment of hPDLCs with 5 μM ML385, a Nrf2-specific inhibitor, suppressed the potency of ASTX to diminish AGE-stimulated ROS accumulation in the cells (Figure 8C,D). In addition, the inhibitory effect of ASTX on the AGE-stimulated increase in MitoSOX-positive hPDLCs was apparently diminished in the presence of ML385 (Figure 8E,F). Overall, the pretreatment with ML385 significantly (*p* < 0.05) inhibited the stimulating effect of ASTX on hPDLC’ proliferation (Figure 8G). Collectively, our results imply that the protective or ameliorating effect of ASTX on diabetes-triggered oxidative stress is closely associated with its ability to upregulate Nrf2 and Nrf2-dependent antioxidant signaling.

## 4. Discussion

Hyperglycemia triggers various degenerative and inflammatory healthy complications. Numerous studies have demonstrated beneficial effects of ASTX on diabetes-associated chronic diseases [16]. Several studies indicate that in addition to anti-inflammatory and antioxidative effects, ASTX attenuates hyperglycemic conditions in diabetic animal models [26]. However, our results show that the supplemental ASTX does not directly reduce blood glucose level in STZ-induced diabetic mice, despite its long-term administration for 60 days. Rather, it is likely that the ASTX treatment not only ameliorates the excessive consumptions of water and feed, but also delays an acute loss of body weight along with improved survival rate in STZ-induced diabetic mice. These findings are similar to a previous report showing that supplemental ASTX does not induce a significant effect on blood glucose level, but inhibits retinal oxidative stress and the expression of inflammatory mediators in STZ-induced diabetic rats [27]. Although the effects of ASTX on diabetes-associated healthy complications may differ from the experimental conditions, our findings indicate that the STZ-induced disruption of pancreatic islets and the resulting hyperglycemia are beyond the capacity that is recoverable by supplemental ASTX. This also implies that ASTX treatment does not improve insulin secretion in type I diabetes patients. Furthermore, our results together with previous reports suggest that the potentials of ASTX to improve water and feed consumption, body weight loss, and lethal rates in STZ-injected mice are mainly derived from its antioxidant ability to ameliorate hyperglycemia-triggered chronic and systemic complications rather than to directly protect against pancreatic cell injury [16,17,18,19,20,21,27].

Periodontitis is one of the potential complications of diabetes [5,6,7,8]. Prolonged periodontitis causes severe resorption of alveolar bone and removal of teeth from the gums. The increases in ROS accumulation and inflammatory response in the periodontium are considered as the key mediators of periodontal disease [28]. Chronic oxidative stress and inflammation evokes an imbalanced bone metabolism and increases the risk of bone fracture in type 1 diabetic patients [29]. Similar to the previous report [15], the current findings support that STZ-mediated hyperglycemia induces alveolar bone loss and connective tissue degradation in the periodontium. Our results also confirm that hyperglycemia directly contributes to periodontitis and/or increases the risk of periodontal diseases.

Interfered bone formation and bone mass loss are the characteristic phenotypes in diabetic complications [30,31]. Studies indicate that dysregulated osteogenic activation in the periodontium participates in diabetes-associated alveolar bone resorption [30,31,32,33]. The cycle of bone growth and resorption is tightly affected by the balanced activation between osteoclasts and osteoblasts. The activation of these cells is regulated by a broad array of hormonal and biological factors. Of them, oxidative stress is the preliminary source causing an imbalanced bone metabolism and leading to osteoclastic activation and inflammatory response [34,35]. Diabetic healthy complications also involve chronic oxidative stress and inflammatory conditions as hyperglycemia-associated characteristics [2,9,12]. As proven by the excessive osteoclastic activation, greater expression of RANKL and CTSK, the increases in 8-OHdG, γ-H2AX, COX-2, and IL-1β levels, and the decreases in RUNX2 and osterix levels in the periodontium of STZ-injected mice, all of our findings strongly suggest that diabetes-triggered periodontal destruction is the outcome orchestrated by imbalanced bone metabolism, oxidative stress, and inflammatory responses. Alternatively, these results indicate that ASTX protects diabetes-triggered periodontal destruction via antioxidative, anti-osteoclastogenic, and anti-inflammatory potentials.

Considerable evidence highlights that Nrf2 and Nrf2-related signaling play key roles in protecting diabetes-associated chronic complications [36]. Nrf2 regulates the expression of multiple antioxidant enzymes that play crucial roles in cellular defenses against inflammatory- and oxidative stress-induced chronic diseases [37,38]. Nrf2 blocks the transcriptional activation of pro-inflammatory cytokines [39]. Nrf2 and Nrf2-dependent pathways are also the main target in ASTX-mediated protection of oxidative and inflammatory chronic complications [2,21,40]. Our findings support the relationship between the Nrf2 level and diabetes-triggered periodontal destruction in STZ-injected mice. Our results also indicate that in parallel with Nrf2, the activities of catalase and SOD are correlated with STZ-induced oxidative complication and its protection by supplemental ASTX. As the hyperglycemia accumulates AGEs that lead to the release of pro-inflammatory mediators in the periodontium of diabetic patients [28,41], our in vitro experiments support the roles of Nrf2 on ASTX-induced suppression of AGE-induced oxidative stress and growth inhibition in hPDLCs.

Otherwise, oxidative stress can disrupt hematopoietic development. Regarding chronic oxidative stress and inflammation, long-term diabetes also leads to impaired repopulation of hematopoietic progenitor cells and dysregulates the production of circulating blood cells [42,43]. The lack of lymphocytes in peripheral blood and/or imbalanced composition of circulating blood cells may increase lethality along with an impaired immune response induction. Our results indicate that the increased lifespan of STZ-injected mice by supplemental ASTX is associated with the recovery of circulating lymphocytes. There is a report showing that ASTX improves irradiation-induced hematopoietic impairments by inhibiting oxidative stress and apoptosis [44]. ASTX also ameliorated imbalanced redox statues in lymphocytes of alloxan-induced diabetic rats [22]. Moreover, it is suggested that Nrf2 is the redox and metabolic sensor regulating stem cell self-renewal, proliferation, and differentiation [45]. Collectively, our study suggests that supplemental ASTX increases the survival rate of STZ-injected mice by ameliorating an impaired hematopoietic development. Further experiments to verify how ASTX affects hematopoietic development in the diabetic animal model are necessary.

## 5. Conclusions

Hyperglycemia evokes chronic oxidative stress and inflammatory responses in the body, through which periodontitis and periodontic degradation is triggered. Here, we demonstrate that supplemental ASTX protects hyperglycemia-triggered periodontal degradation and ameliorates hyperglycemia-associated systemic complications in STZ-induced type I diabetic mice. Our results indicate that the protective potentials of ASTX on STZ-mediated diabetic complications are not directly associated with its ability to improve hyperglycemic conditions, as well as to suppress pancreatic islets injury. Rather, our findings support that a recovery of Nrf2-regulated antioxidant systems and the attendant suppression of oxidative damage, osteoclastic activation, and inflammatory responses are the main mechanisms by which ASTX inhibits diabetes-triggered degenerative complications. Overall, this study provides a possibility that supplemental ASTX improves and/or ameliorates periodontal destruction and oxidative systemic complications in type I diabetic patients.

## Figures and Tables

**Figure 1 nutrients-13-03575-f001:**
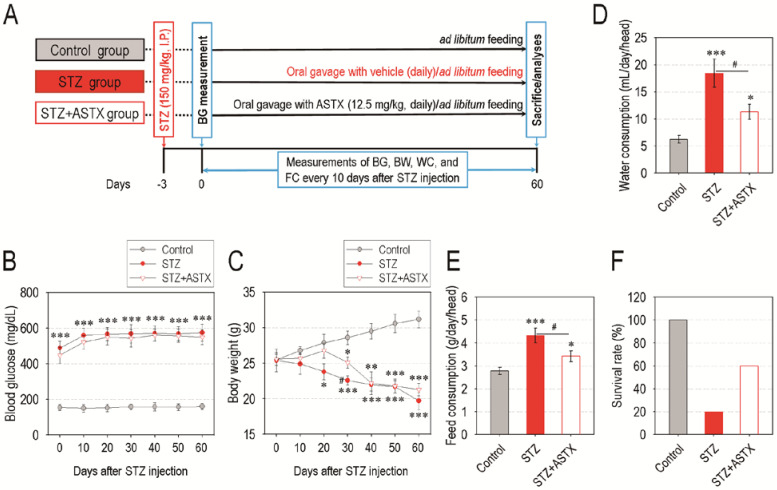
Oral gavage with astaxanthin (ASTX) does not ameliorate blood glucose level, but decreases excessive consumptions of water and feed and lethality in STZ-induced diabetic mice. (**A**) A scheme illustrating the experimental designs. (**B**) Blood glucose level and (**C**) body weight in mice groups were measured at the indicated days after hyperglycemia induction (*n* = 10). Amounts of (**D**) water and (**E**) feed consumption in mice groups were evaluated 30 days after hyperglycemia induction (*n* = 7). (**F**) Survival rate (%) of mice groups 12 h after the oral gavage for 60 days with ASTX (*n* = 10). * *p <* 0.05, ** *p <* 0.01, and *** *p <* 0.001 vs. control group; ^#^
*p* < 0.05 vs. streptozotocin (STZ) group. BG, blood glucose; BW, body weight; WC, water consumption; FC, feed consumption.

**Figure 2 nutrients-13-03575-f002:**
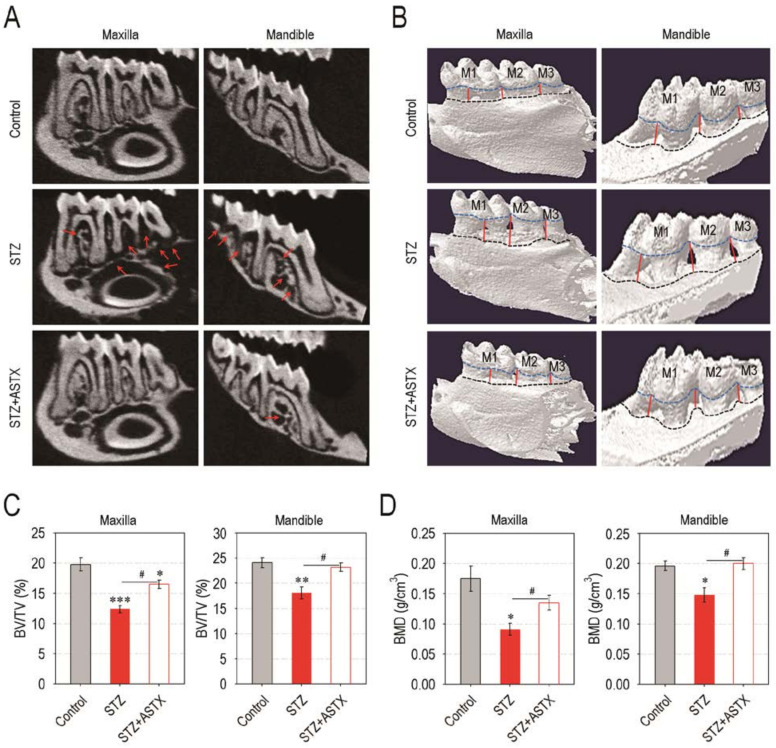
Supplemental ASTX restores hyperglycemia-triggered destruction of alveolar bone in STZ-induced diabetic mice. (**A**) Representative 2D µCT images showing alveolar bone around the molars in the maxilla and the mandible 60 days after hyperglycemia induction. The red arrows indicate the resorbed alveolar bones around the upper and lower molars. A representative result from five different samples is shown. (**B**) The reconstructed 3D µCT images of the maxilla and the mandible. In 3D images, blue and black dash lines indicate CEJ and ABC, respectively, while red lines represent a boundary of resorbed alveolar bone in the lower and upper molar regions. (**C**) The values of bone volume/tissue volume (BV/TV, %) and (**D**) bone mineral density (BMD, g/cm^3^) in the maxilla and the mandible were calculated from the μCT images (*n* = 5). * *p <* 0.05, ** *p <* 0.01, and *** *p <* 0.001 vs. control group; ^#^
*p* < 0.05 vs. STZ group.

**Figure 3 nutrients-13-03575-f003:**
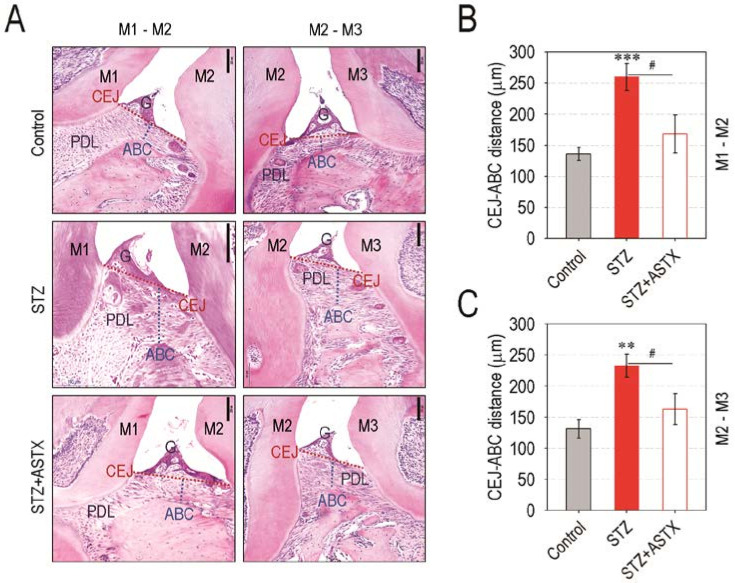
ASTX administration suppresses periodontal tissue degradation in STZ-induced diabetic mice. (**A**) H & E staining exhibiting hyperglycemia-triggered periodontal degradation and its suppression by supplemental ASTX in the interproximal and furcation region of maxillary molars (M1 to M3) in STZ-injected mice 60 days after hyperglycemia induction (100× magnification). G, gingiva. Bar = 200 μm. The distance (μm) between CEJ and ABC in the maxillary regions of (**B**) M1 to M2 and (**C**) M2 to M3 was calculated using ImageJ program (*n* = 5). ** *p <* 0.01 and *** *p <* 0.001 vs. control group; ^#^
*p* < 0.05 vs. STZ group.

**Figure 4 nutrients-13-03575-f004:**
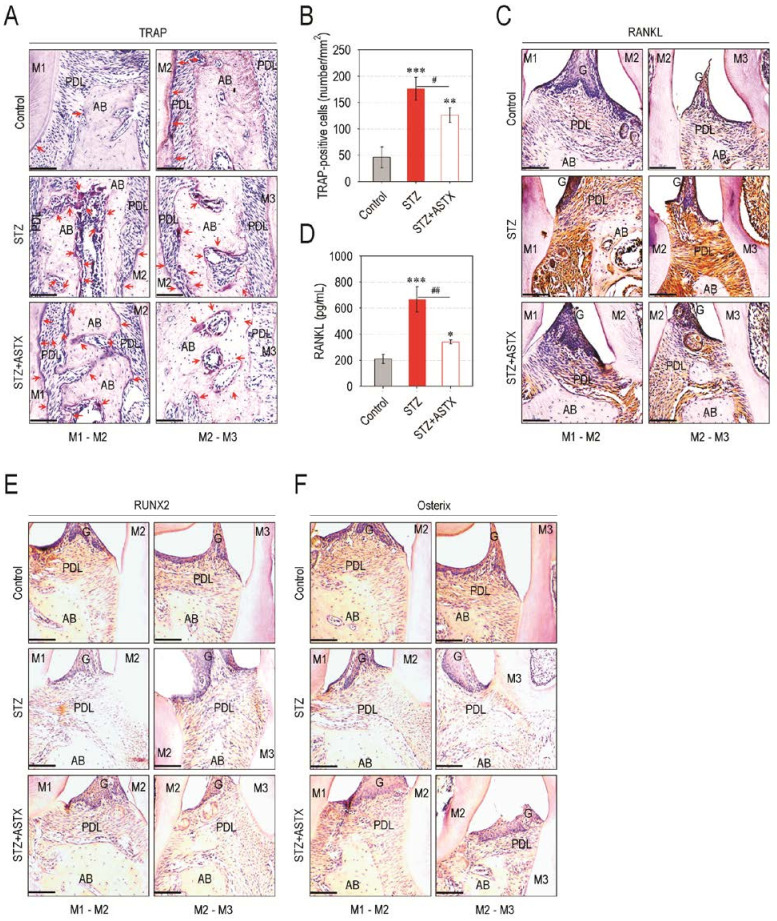
Oral gavage with ASTX restores hyperglycemia-associated imbalanced activation between osteoclasts and osteoblasts in STZ-induced diabetic mice. (**A**) TRAP staining showing osteoclast formation in the interproximal region of maxillary molars (M1 to M3) at 60 days post-hyperglycemia induction (200× magnification). Bar = 100 μm. The red arrows indicate TRAP-positive osteoclasts. (**B**) Area of the TRAP-positive cells (mm^2^) was determined using ImageJ program (*n* = 6). (**C**) IHC staining exhibiting the expression level of RANKL at the region of upper molars (200× magnification). Bar = 100 μm. (**D**) The level of RANKL in blood serum of mice groups was determined by ELISA 60 days after hyperglycemia induction (*n* = 6). Levels of (**E**) RUNX2 and (**F**) osterix in the maxilla were also evaluated by IHC at the same time (200× magnification). Bar = 100 μm. * *p <* 0.05, ** *p <* 0.01, and *** *p <* 0.001 vs. control group; ^#^
*p* < 0.05 and ^##^
*p* < 0.01 vs. STZ group.

**Figure 5 nutrients-13-03575-f005:**
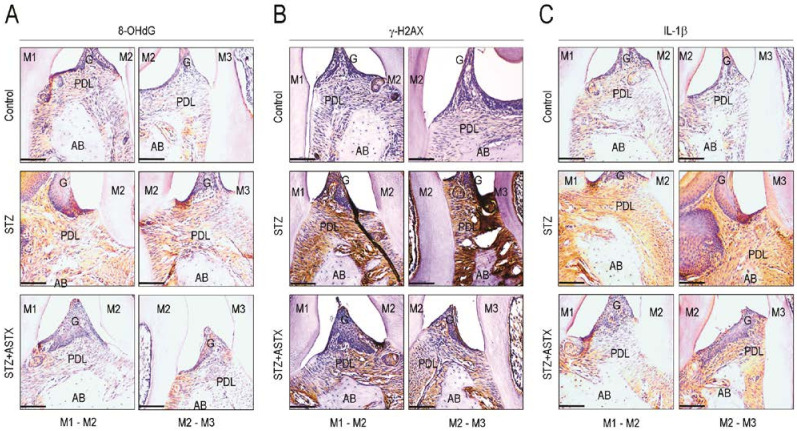
Supplemental ASTX inhibits oxidative stress, DNA damage, and expression of inflammatory cytokines in the periodontium of STZ-induced diabetic mice. Expression levels of (**A**) 8-OHdG, (**B**) γ-H2AX, and (**C**) IL-1β in periodontal tissue sections including the region of maxillary molars (M1 to M3) were evaluated by IHC assay using their specific antibodies 60 days after hyperglycemia induction (200× magnification). Bar = 100 μm. A representative result from five different samples per each experiment is shown.

**Figure 6 nutrients-13-03575-f006:**
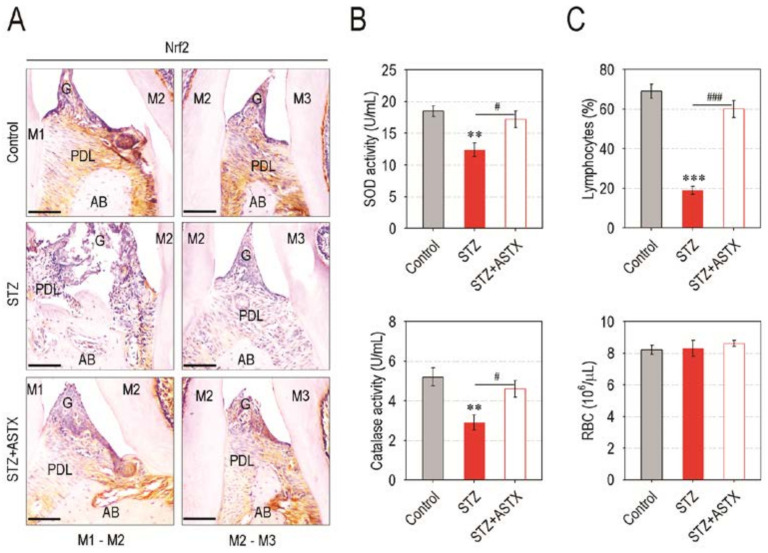
ASTX treatment restores the levels of Nrf2, antioxidant enzymes, and circulating lymphocytes in STZ-induced diabetic mice. (**A**) Expression level of Nrf2 in the region of upper molars of mice groups were determined by IHC 60 days after hyperglycemia induction (200× magnification). Bar = 100 μm. (**B**) The activities of SOD and catalase in blood serum of mice groups were examined using their specific assay kits (*n* = 5). (**C**) Circulating levels of lymphocytes and RBC in mice groups were calculated using a blood cell counter (*n* = 5). The enzyme activities and blood cell numbers were checked 60 days after hyperglycemia induction. ** *p <* 0.01 and *** *p <* 0.001 vs. control group; ^#^
*p* < 0.05 and ^###^
*p* < 0.001 vs. STZ group.

**Figure 7 nutrients-13-03575-f007:**
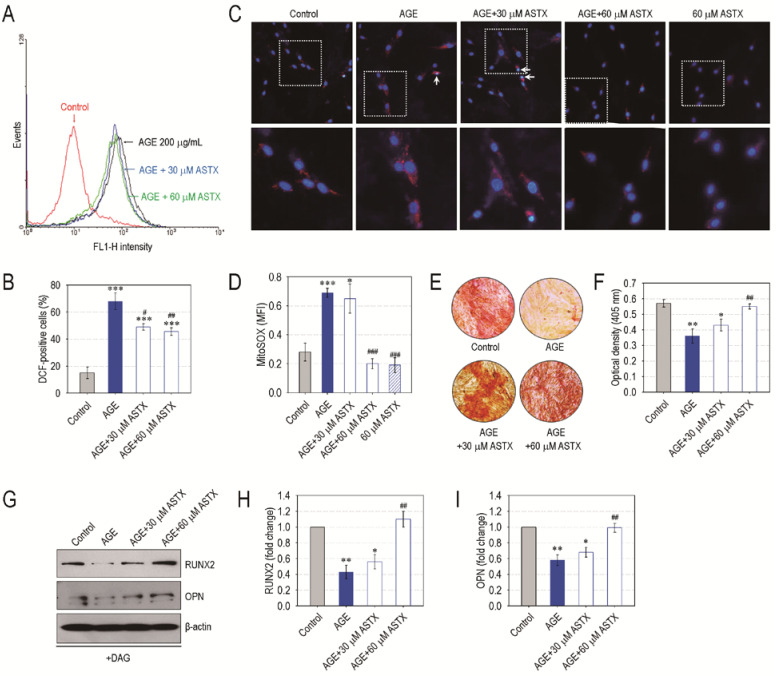
ASTX inhibits oxidative stress and increases mineralization in AGE-treated hPDLCs. The hPDLCs were incubated in growth medium supplemented with and without AGE (200 μg/mL), ASTX (30 or 60 μM), or both for 48 h. (**A**) ROS accumulation in the cells was determined by flow cytometry using DCFH-DA, and (**B**) the DCF-positive cells (%) were calculated (*n* = 5). (**C**) The level of mitochondrial superoxide in the cells was determined by staining them with MitoSOX Red and DAPI, and (**D**) the MFI specific to the red dye was calculated using ImageJ software (*n* = 5). Alternatively, the hPDLCs were incubated in DAG-supplemented osteogenic medium in the presence and absence of AGE, ASTX or both. (**E**) After 14 days of incubation, the cells were stained with Alizarin Red S, and (**F**) the red dye-specific optical density in the cells was determined using a microplate reader (*n* = 5). (**G**) The protein levels of osteogenic markers in the cells exposed to AGE, ASTX, or both were analyzed by immunoblot assay 5 days after incubation, and the band intensities of (**H**) RUNX2 and (**I**) OPN were calculated (*n* = 4). * *p* < 0.05, ** *p* < 0.01, and *** *p* < 0.001 vs. the untreated control hPDLCs. ^#^
*p* < 0.05, ^##^
*p* < 0.01, and ^###^
*p* < 0.001 vs. the hPDLCs exposed to AGE alone.

**Figure 8 nutrients-13-03575-f008:**
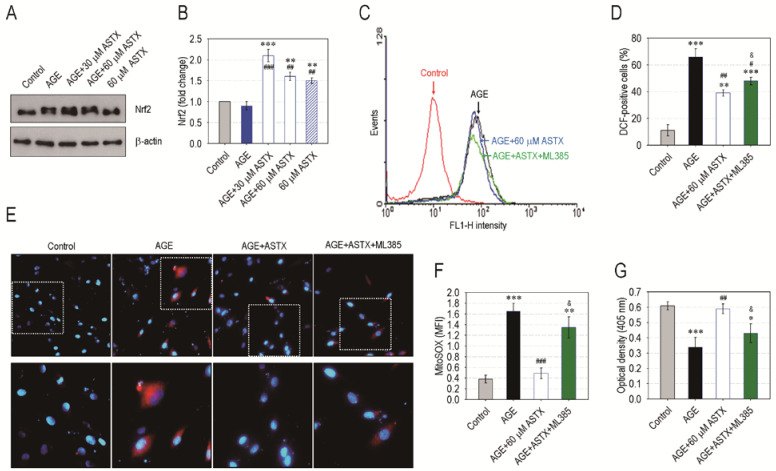
ASTX inhibits oxidative stress and increases proliferation in AGE-exposed hPDLCs by upregulating Nrf2 pathway. The hPDLCs were incubated in growth medium supplemented with and without AGE (200 μg/mL), ASTX (30 or 60 μM), or both. (**A**) After 48 h of incubation, the protein level of Nrf2 in the cells was determined by immunoblot assay, and (**B**) the Nrf2-specific band intensity (fold change to control) was calculated after normalizing its level to that of β-actin (*n* = 4). The hPDLCs were also exposed to AGE, ASTX, or both with and without the pretreatment with 5 μM ML385. After 48 h of incubation, (**C**) the DCF-specific signal and (**D**) the DCF-positive cells (%) in the hPDLCs were calculated by flow cytometry (*n* = 5). After the same time, (**E**) immunofluorescence assay was also performed using MitoSOX Red and DAPI, and (**F**) the red dye-specific MFI among the experiments was compared after calculating the levels using ImageJ software (*n* = 5). (**G**) In addition, proliferation rate of the cells was determined at the end of 48 h incubation using a CCK-8 assay kit. * *p* < 0.05, ** *p* < 0.01, and *** *p* < 0.001 vs. the untreated control hPDLCs. ^#^
*p* < 0.05, ^##^
*p* < 0.01, and ^###^
*p* < 0.001 vs. the hPDLCs exposed to AGE alone. ^&^
*p* < 0.05 vs. the cells exposed to ASTX in combination with AGE.

## Data Availability

The data used to support the findings of this study are available from the corresponding authors upon request.

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
