# Peer review of "Astaxanthin Inhibits Diabetes-Triggered Periodontal Destruction, Ameliorates Oxidative Complications in STZ-Injected Mice, and Recovers Nrf2-Dependent Antioxidant System"

_nutrients, 2021, doi:10.3390/nu13103575_

Round 1
Reviewer 1 Report
- Line 41. “utilization” should better read “action”
- Line 48. “complicative diseases” should better read “complications”
- Line 62. “clinical use of non-toxic and antioxidative dietary nutrients is to be the attractive approach”
- Figure 1D, 1E. It is not clear how water and feed consumption was assessed. The authors should describe in the methods section and/or in the figure legend if the mice were single-housed and for how long in order to perform these measurements. Did they use any special equipment to measure water and food consumption so as to account for the losses in the regular cages?
- Figure 1F. It is not clear when the mice died. Some of them died before the final dose of the drug? If this is the case, some of the tissue sections are collected post-mortem?
- Line 495. “Even though” had better read “despite”.
- Line 509. “protect pancreatic cell injury” had better read “protect against pancreatic cell injury”
- Figure 5. Legends should describe in details all the panels of a figure.
- Figure 5A, 5B. It seems that there are increased levels of oxidative stress biomarker and marker of DNA damage in the mice treated with STZ. Figure 6 shows that Nrf2 protein is decreased in mice with STZ treatment. This is an interesting finding. One would expect that the hyperglycemia-induced oxidative stress would lead to Nrf2 pathway activation. SOD activity and catalase activity are reduced and this could lead us to the possible explanation that the suppression of these enzymes leads to increased oxidative stress. However, it would be more informative and more supportive of the data if the authors measured at the RNA or protein level a more prototypical Nrf2 target gene such as Nqo1, Gclc or similar. SOD and catalase are partially regulated by Nrf2, that’s why I suggest that this experiment would be more informative.
- Figure 8A, 8B. In similar way with Figure 5A, 5B, measurement at the mRNA or protein level of Nqo1, Gclc as a more prototypical Nrf2 target would be informative. These targets should be measured in the ASTX samples as well.
- As the authors do not employ Nrf2 knockout mouse models, it is not totally clear if the effects of ASTX are mediated by Nrf2 partially or completely. Therefore, the title had better become a bit more descriptive so as to depict this fact. “Astaxanthin Inhibits Diabetes-Triggered Periodontal Destruction, Ameliorates Oxidative Complications in STZ-Injected Mice and RecoversNrf2-Dependent Antioxidant System”.
Author Response
Comments and Suggestions for Authors (Reviewer #1)
- Line 41. “utilization” should better read “action”
► Author response : Thank you for the helpful comment. The authors revised the word in the revised manuscript.
- Line 48. “complicative diseases” should better read “complications”
► Author response : Thank you for the helpful comment. The authors revised the words in the revised manuscript.
- Line 62. “clinical use of non-toxic and antioxidative dietary nutrients is to be the attractive approach”
► Author response : Thank you for the helpful comment. The authors revised the sentence in the revised manuscript.
- Figure 1D, 1E. It is not clear how water and feed consumption was assessed. The authors should describe in the methods section and/or in the figure legend if the mice were single-housed and for how long in order to perform these measurements. Did they use any special equipment to measure water and food consumption so as to account for the losses in the regular cages?
► Author response : We understood on the reviewers’ comments. To further clarify how the water and feed consumptions were measured, we provided additional methods into the section of 2.2 in the revised manuscript.
- Figure 1F. It is not clear when the mice died. Some of them died before the final dose of the drug? If this is the case, some of the tissue sections are collected post-mortem?
► Author response : Yes, the STZ-injected mice exhibited a time-dependent lethality during the experimental period. To adjust the numbers among the groups, we prepared 2- or 3-fold STZ and STZ+ASTX groups more than the control group. All tissue samples used in this study were collected at the final gavage of ASTX, namely after 12 h of the gavage for 60 days. The authors hope that this explanation resolves the reviewers’ question.
- Line 495. “Even though” had better read “despite”.
► Author response : Thank you for the helpful comment. The authors revised the word in the revised manuscript.
- Line 509. “protect pancreatic cell injury” had better read “protect against pancreatic cell injury”
► Author response : Thank you for the helpful comment. The authors revised the words in the revised manuscript.
- Figure 5. Legends should describe in details all the panels of a figure.
► Author response : Thank you for the notification. This may be derived from the editorial edition to change the form of originally submitted manuscript. We added the figure legends.
- Figure 5A, 5B. It seems that there are increased levels of oxidative stress biomarker and marker of DNA damage in the mice treated with STZ. Figure 6 shows that Nrf2 protein is decreased in mice with STZ treatment. This is an interesting finding. One would expect that the hyperglycemia-induced oxidative stress would lead to Nrf2 pathway activation. SOD activity and catalase activity are reduced and this could lead us to the possible explanation that the suppression of these enzymes leads to increased oxidative stress. However, it would be more informative and more supportive of the data if the authors measured at the RNA or protein level a more prototypical Nrf2 target gene such as Nqo1, Gclc or similar. SOD and catalase are partially regulated by Nrf2, that’s why I suggest that this experiment would be more informative.
► Author response : Thank you for the critical and helpful comments and the authors deeply agreed with the reviewers’ comments for the further informative and supportive results. Although some data in the current manuscript are able to support the role of Nrf2, we also consider that there was still the lack of data to clearly support the mechanisms by which Nrf2 or Nrf2-related pathway regulates oxidative stress and the cellular redox regulatory systems in STZ and STZ+ASTX groups. In addition to the evaluation of Nrf2 target genes in our experimental model, we consider that using the transgenic Nrf2 and diabetes animal models might provide further detail mechanisms and impacts of ASTX on type I diabetes. Notwithstanding, we carefully respond that these additional works will be in future studies.
- Figure 8A, 8B. In similar way with Figure 5A, 5B, measurement at the mRNA or protein level of Nqo1, Gclc as a more prototypical Nrf2 target would be informative. These targets should be measured in the ASTX samples as well.
► Author response : About these comments, please consider the authors response described above.
- As the authors do not employ Nrf2 knockout mouse models, it is not totally clear if the effects of ASTX are mediated by Nrf2 partially or completely. Therefore, the title had better become a bit more descriptive so as to depict this fact. “Astaxanthin Inhibits Diabetes-Triggered Periodontal Destruction, Ameliorates Oxidative Complications in STZ-Injected Mice and RecoversNrf2-Dependent Antioxidant System”.
► Author response : The authors fully agreed with the reviewers’ comments. Based on the recommendation, we revised the title in the revised manuscript.
Reviewer 2 Report
The manuscript is well written and it was well organized. I have few comments to improve the clarity of the manuscript
- What is the effect of olive oil over STZ induced diabetic mice?
- How the dosage of 12.5 mg/kg ASTX was fixed?
- Kindly rewrite the sentence in line 65 to 68.
- How ASTX ameliorates the excessive consumption of water and feed and delays the loss of body weight? Explain in detail.
Author Response
Comments and Suggestions for Authors (Reviewer #2)
The manuscript is well written and it was well organized. I have few comments to improve the clarity of the manuscript
- What is the effect of olive oil over STZ induced diabetic mice?
► Author response : As many investigators have used and ASTX is not hydrophilic, we also selected the olive oil as vehicle. Considering other reports along with our previous studies, olive oil itself does not affect diabetic complications and any side effects in animal models. To provide an additional understanding on the use of olive oil in diabetic models, we added a sentence together with the related references into the section of 2.2. in the revised manuscript.
- How the dosage of 12.5 mg/kg ASTX was fixed?
► Author response : Similar to the authors’ response in the comment 1, we considered numerous studies to select the dosage of ASTX in STZ-injected diabetic mice. In addition to the related references that were provided in the section of 2.2. in the revised manuscript, there are also many studies to consider the dosage of ASTX, in which from 5 to 100 mg/kg of ASTX along with different supplementation periods (from 7 to 30 days) were applied in diabetic animal models. Collectively, considering the administration period of ASTX, we finally selected 12.5 mg/kg of ASTX as the minimal dosage to exert biological effects.
- Kindly rewrite the sentence in line 65 to 68.
► Author response : We carefully checked the sentence and revised adequately them in the revised manuscript.
- How ASTX ameliorates the excessive consumption of water and feed and delays the loss of body weight? Explain in detail.
► Author response : As the authors suggested in Discussion section, we consider that the potentials of ASTX to improve water and feed consumption, body weight loss, and lethal rate in STZ-injected mice are mainly derived from its antioxidant ability to ameliorate hyperglycemia-triggered chronic and systemic complications rather than to directly protect against pancreatic cell injury. To further clarify that, we revised the related sentence in Discussion section of the revised manuscript as represented above.